# New Insights on the Emerging Genomic Landscape of CXCR4 in Cancer: A Lesson from WHIM

**DOI:** 10.3390/vaccines8020164

**Published:** 2020-04-03

**Authors:** Stefania Scala, Crescenzo D’Alterio, Samantha Milanesi, Alessandra Castagna, Roberta Carriero, Floriana Maria Farina, Massimo Locati, Elena Monica Borroni

**Affiliations:** 1Functional Genomics, Istituto Nazionale Tumori -IRCCS- Fondazione G. Pascale, via M. Semmola, 80131 Napoli, Italy; 2Department of Medical Biotechnologies and Translational Medicine, Università degli Studi di Milano, via Fratelli Cervi 93, 20090 Milan, Italy; 3Humanitas Clinical and Research Center—IRCCS, via Manzoni 56, 20089 Rozzano (Mi), Italy; 4Bioinformatic Unit, Humanitas Clinical and Research Center—IRCCS, via Manzoni 56, 20089 Rozzano (Mi), Italy

**Keywords:** cancer, CXCR4, mutations, WHIM

## Abstract

Deciphering the molecular alterations leading to disease initiation and progression is currently crucial to identify the most relevant targets for precision therapy in cancer patients. Cancers express a complex chemokine network influencing leucocyte infiltration and angiogenesis. Moreover, malignant cells also express a selective repertoire of chemokine receptors that sustain their growth and spread. At present, different cancer types have been shown to overexpress C-X-C chemokine receptor type 4 (CXCR4) and to respond to its ligand C-X-C motif chemokine 12 (CXCL12). The CXCL12/CXCR4 axis influences cancer biology, promoting survival, proliferation, and angiogenesis, and plays a pivotal role in directing migration of cancer cells to sites of metastases, making it a prognostic marker and a therapeutic target. More recently, mutations in the C-terminus of CXCR4 have been identified in the genomic landscape of patients affected by Waldenstrom’s macroglobulinemia, a rare B cell neoplasm. These mutations closely resemble those occurring in Warts, Hypogammaglobulinemia, Immunodeficiency, and Myelokathexis (WHIM) syndrome, an immunodeficiency associated with CXCR4 aberrant expression and activity and with chemotherapy resistance in clinical trials. In this review, we summarize the current knowledge on the relevance of CXCR4 mutations in cancer biology, focusing on its importance as predictors of clinical presentation and response to therapy.

## 1. CXCL12/CXCR4 Axis in Tumors

### 1.1. CXCL12

The tumor microenvironment (TME) is a dynamic heterocellular niche that affects cancer fate. Critical interactions occurring in the TME mostly promote tumor initiation, resistance, metastasis, and recurrence [1]. The CXCL12/CXCR4 axis bonds cancer cells and the TME [2]. Chemokines are small molecules belonging to a superfamily of chemo-attractive cytokines [3]. Chemokines are grouped into four subfamilies—CC, XC, CXC, and CX3C (where C are the two N-terminal cysteines and X represents any aminoacid)—and are functionally classified into two groups—homeostatic and inflammatory subtypes [3]. The homeostatic group of chemokines is formed constitutively, while the inflammatory group is inducible and is involved in recruitment of immune cells toward the areas of inflammation [3]. The naming of chemokine receptors is depended on their respective chemokine ligands. There are two types in this category—conventional G protein-coupled and atypical G protein-uncoupled chemokine receptors [3]. Among a number of chemokines and their receptors, CXCL12/CXCR4 signaling is an integral oncogenic communication network between a tumor and its stroma [3].

CXCL12 (also known as SDF-1) is a chemokine belonging to the CXC subfamily that was cloned from bone narrow (BM)-derived stromal cell line and later identified as a growth stimulation factor for pre-B cells [4]. Widely secreted in different tissues by stromal cells, fibroblasts, and epithelial cells, it binds to the extracellular matrix through a cluster of basic residues—the BBXB motif (B for basic amino acid and X any amino acid) [5]. CXCL12 binds the seven α-helical transmembrane domains (7TM), G protein-coupled (GPCR) conventional receptor CXCR4 and Atypical ChemoKine Receptor 3 (ACKR3, also known as CXCR7), and plays a key role in physiological and pathological processes, including embryogenesis, hematopoiesis, angiogenesis, and inflammation, regulating the migration of hematopoietic progenitor and stem cells, endothelial cells, and leukocytes [6,7,8]. CXCR4 is expressed in a great variety of cell types, including lymphocytes, hematopoietic stem cells, endothelial cells, epithelial cells, stromal fibroblasts, and cancer cells. CXCR4-positive tumor cells migrate along the CXCL12 gradient to distant organs, such as the CXCL12-rich BM microenvironment [8]. CXCL12 not only induces migration but also regulates adhesion of tumor cells with laminin, fibrinogen, stromal cells, and endothelial cells by activating cell surface adhesion molecules in pancreatic [9], prostate [10,11], ovarian [12], and small cell lung cancer (SCLC) cells [13]. CXCL12 regulates invasion in melanoma and colorectal cancer [14] and myeloma [15]. In a meta-analysis of 38 studies involving 5807 patients, high CXCL12 expression was associated with reduced overall survival in patients with esophagogastric (Hazard Ratio (HR) 2.08; 95% Confidence Interval (CI): 1.31–3.33, *p* = 0.002), pancreatic (HR 1.54; 95% CI: 1.21–1.97, *p* = 0.0005), and lung cancer (HR 1.37; 95% CI: 1.08–1.75, *p* = 0.01), whereas in breast cancer patients, high CXCL12 expression conferred an overall survival advantage (HR 0.5; 95% CI: 0.38–0.66, *p* < 0.00001) [16]. CXCL12 exists in six different splice variants in humans (α, β, γ, δ, ε, θ) and three (CXCL12α to γ) in mice [17,18]. CXCL12-α is not present in blood due to enzymatic degradation [19], but it is expressed in adult BM, which is responsible for progenitor cell retention and chemotaxis of leukemia cells [20]. Unlike CXCL12-α, CXCL12-β promotes angiogenesis [21], while CXCL12-γ is highly expressed in less vascularized organs, such as the heart and brain [22]. CXCL12-γ induces weak chemotaxis in vitro, while in vivo it is the most active in stimulating chemotaxis [5]. Due to a stable binding interaction, CXCL12-γ produces an inflammatory reaction that is more efficient than the α-isoform in in vivo mouse models [19]. CXCL12-γ has been proposed as a prognostic marker for breast cancer [22]. CXCL12-γ mostly expressed by carcinoma-associated fibroblasts confer to CXCR4-positive breast cancer cells the ability to metastasize into BM through the expression of the receptor activator of NFκB ligand (RANKL) [21]. CXCL12-γ overexpression induces cancer stem cells and neuroendocrine phenotypes in prostate cancer cells through activation of PKCα/NFκB-dependent signaling [23].

### 1.2. CXCR4

Encoded on chromosome 2q21, CXCR4 is an evolutionarily highly conserved GPCR identified as the leucocyte-expressed 7TM receptor (LESTR)/fusin, a co-receptor for HIV [24]. The binding with CXCL12 activates the proinflammatory signaling pathway nuclear factor kappa-light-chain-enhancer of activated B cells (NFκB), the Janus kinase/signal transducers and activators of transcription (JAK/STAT), Phosphatidylinositol-3-kinase (PI3K)/ Ser and Thr kinase AKT, also known as protein kinase B (PKB) pathways, and mammalian target of rapamycin protein kinase (mTOR). The Jun N-terminal kinase (JNK) and p38 mitogen-activated protein kinase (MAPKs) signaling trigger cell survival, proliferation, and chemotaxis of both immune and cancer cells [25]. At present, 23 different cancer types have been shown to overexpress CXCR4 and to respond to its ligand CXCL12, including breast cancer, ovarian cancer, small-cell lung cancer, hepatocellular carcinoma, gastric and stomach cancers, melanoma, brain tumors, soft tissue sarcomas, pancreatic cancer, brain cancer, and ovarian cancer [25,26]. CXCR4 can be activated in cancer cells through the following mechanisms: (i) hypoxia upregulates the CXCL12–CXCR4 axis [27]; (ii) CXCR4 regulates stem cell pluripotency and cell fate decisions through Wnt/β-catenin, which in turn positively modulates CXCR4 expression [25,28]; (iii) NFκB induces CXCR4 expression and stimulates tumor invasion [29]. CXCR4 promoter activity is upregulated by c-Myc and downregulated by Yin-Yang [30]. Post-transcriptional regulation of CXCR4 was reported in breast cancer, where the oncogene Her2 blocks its ubiquitination and degradation [31]. The CXCL12–CXCR4 signaling pathway transactivates Her2-neu to stimulate invasive and metastatic signals in breast [32], esophageal [33], lung [34], and prostate [35] cancers, as well as ovarian cancers [36]. Recent evidence has shown CXCR4-dependent mTOR signaling in pancreatic cancer, gastric cancer, and T-cell leukemia cells [25]; in fact, CXCR4 and mTOR inhibitors have been reported to impair human renal cancer migration [37]. Given the undisputed clinical relevance of the CXCL12/CXCR4 axis, a multitude of antagonists have been developed. A CXCL12 mimetic cyclic peptide [38] potentiates anti- Programmed death-ligand 1 receptor-1 (PD-1) efficacy in vivo [39] and reverts the suppressive activity of T regulatory cells in renal cancer ex vivo [40]. A possible mechanistic explanation involves the demethylation of Forkhead bOX P3 FOXP3 promoter in T regulatory cells (Treg) [41]. CXCL12 also binds CXCR7 with high affinity [42] and may induce CXCR7 internalization without the canonical Gi-mediated signaling [43]. CXCL11/I-TAC also binds CXCR7 [42]. Similar to CXCR4, CXCR7 is also widely detected in the central nervous systems, neural stem cells, liver oval and stem cells, CD34+ hematopoietic progenitor cells, white blood cells primordial germ cells, skeletal muscle satellite progenitor cells, as well as the intestinal epithelium [30]. CXCR7 forms functional heterodimers with CXCR4, allowing Gi activation, calcium responses, and chemotaxis in T lymphocytes [44]. The CXCR7/CXCR4 heterodimer can recruit nuclear β-arrestin 1 to enhance cancer cell migration and activate PI3K and MAPK pathways [45]. It was recently shown that the β-arrestin 1 recruitment to the nucleus increases histone demethylase JMJD2A (jumonji (jmj) domain containing protein 2A. Demethylation of histones H3K9 (Histone H3 Lysine 9) and H3K36 (Histone H3 Lysine 36) activates the transcription of pro-inflammatory genes and oncogenes in vivo Granulocyte/Macrophage Colony Stimulating Factor (GM-GSF), Macrophage Colony-Stimulating Factor (M-CSF), InterLeukin 6 (IL-6), c-Myc, Tumor Necrosis Factor alpha (TNF-alpha) in colitis-associated cancer mouse models. Knockdown of βarr1 or inhibition of histone lysine demethylase JMJD2A inhibits colitis and colitis-driven tumorigenesis [46]. Further CXCR7/CXCR4 heterodimers increase the infiltration of T regulatory cells, myeloid-derived suppressor cells, and M2-like macrophages in colonic tissues, and also promotes tumorigenesis [46]. CXCR7 could also heterodimerize with Epidermal Growth Factor Receptor (EGFR) [46,47] to activate MAPK pathways [48]; however, the activation of the MAPK signaling by CXCR7 occurs in conditioning of EGFR inhibition with tyrosine kinase inhibitors (TKIs). CXCR7 promotes resistance to the TKI osimertinib in non-small-cell lung cancer (NSCLC) [49]. CXCR7 overexpression has recently been linked to acquired enzalutamide resistance in prostate cancer [50]. CXCR7 is ubiquitously overexpressed in acquired EGFR TKI resistant cell line NSCLC models with an epithelial–mesenchymal transition (EMT) phenotype [49]. While CXCR4 is known to promote EMT, the depletion of CXCR7 alone is sufficient to reverse the EMT phenotype in lung cancer models [49]. The CXCL12/CXCR4 axis is a molecular hub for neo-angiogenesis, as (i) Vascular Endothelial Growth Factor (VEGF) induces expression of CXCR4 in an autocrine manner in cancer cells [51] and (ii) CXCL12 induces endothelial cells to express VEGF, which in turn promotes the expression of CXCL12 in the same cells [52]. Furthermore, CXCL12 promotes angiogenesis through the recruitment of BM-derived stem cells, which differentiate into endothelial cells and fibroblasts to form the granulation tissue [53,54]. Fang et al. showed that miR-622 inhibits colorectal cancer tumorigenesis and metastasis by downregulating CXCR4, thereby reducing Vascular Endothelial Growth Factor A (VEGFA) and CD34, resulting in an antiangiogenic effect [55]. 

The prognostic role of CXCR4 was confirmed in a recent meta-analysis, including a total of 11 relevant articles involving 1439 pancreatic ductal adenocarcinoma (PDAC) patients, indicating CXCR4 as a negative prognostic marker associated with the risk of lymph node involvement and distant metastasis [56]. Furthermore, CXCR4 is a cancer stem cell marker in PDAC [57], prostate cancer [58], and renal cell carcinoma [59]. Another meta-analysis of 25 selected articles involving 3796 patients with colorectal cancer confirmed that CXCR4 expression is related to tumor–node–metastasis (TNM) stage, tumor differentiation, liver metastasis, lymph node metastasis, distant metastasis, and reduced survival [60]. In renal cell carcinoma, a meta-analysis of 1203 patients from 14 eligible studies confirmed that CXCR4 expression is associated with increased risk, progression, and prognosis, and with poor survival [61]. Furthermore, 24 studies including 3637 patients with gastrointestinal cancer suggested that CXCR4 may serve as a prognostic indicator. Subgroup analysis also indicated that high CXCR4 expression in esophagus, gastric, and colorectal cancers was associated a worse prognosis [62]. Interestingly, an exosome biomarker meta-analysis of 921 breast cancer patients from 11 studies showed that CXCR4 expression together with Her2, Kinase insert Domain Receptor (KDR), CD49d, and CD44 characterized the exosomal protein pattern associated with tumor recurrence or distant organ metastasis [63]. In glioblastoma, the poor patient survival is at least partly caused by glioblastoma stem cells (GSCs) that reside in hypoxic and peri-arteriolar protective niches. CXCR4 antagonists are promising as GSC mobilizers and therapy sensitizers, as they induce GSC differentiation into rapidly-dividing progenitor cells that are more vulnerable to chemotherapy and radiotherapy [64]. 

High levels of CXCL12 and CXCR4 were reported in sinusoidal endothelial cells in Hepatocellular Carcinoma (HCC) specimens [65]. The majority of breast tumors express higher levels of CXCR4, with very low expression in normal breast tissues [66]. CXCR4 expression is high in primary and metastatic lung cancers [67]. Bladder cancer cells express CXCR4 expression, while mRNA in normal urothelial cells is almost undetectable [68]. Additionally, brain tumor cell lines, primary tumors, and metastases overexpress CXCR4 [69]. The Oncomine (Life Technologies, Carlsbad, CA, USA) datasets indicate a high expression of CXCR4 mRNA in head and neck squamous cell carcinoma (HNSCC) and glioblastoma, breast, and pancreatic tumors compared to normal tissues [70]. In Figure 1, the CXCR4 mRNA levels in different tumors are investigated by public gene expression, with higher CXCR4 expression being shown in hematological malignancies, such as acute myeloid leukemia (AML) and diffuse large B-Cell lymphoma (DLBCL), followed by thymoma, clear cell renal cell carcinoma (ccRCC), and adrenocortical carcinoma (Figure 1a). In breast carcinoma, brain tumor: glioblastoma, glioma, kidney-related carcinomas, HNSCC, cholangiocarcinoma, stomach, esophageal, and uterine carcinomas, CXCR4 shows significantly higher mRNA expression levels in cancer cells compared to normal counterparts (Figure 1b). In contrast, in bladder, lung squamous, pancreatic, prostate and rectum carcinomas, CXCR4 showed significantly higher mRNA expression levels in normal cells adjacent to tumor tissue compared to cancer cells (Figure 1b). 

In conclusion, the tumors are dynamic tissue masses requiring continuous exposure to the host cells feeding them, which pave a path for tumor growth and metastasis. CXCL12/CXCR4 is the key signaling used to attract and cross-connect multiple cells within the TME aiming for tumor progression and metastasis. BM, lymph nodes, lung, and liver are common sites of metastasis producing high expression of CXCL12. CXCR4 activation induces tumor angiogenesis, inflammation, and cellular reprogramming, contributing to metastasis. Multiple cells in the TME are recruited by CXCL12 and multiple cells in the TME release CXCL12. TME conditions such as hypoxia act as inducers of CXCR4/CXCL12. Cross-talking between CXCL12/CXCR4 with signals, including NF-kB, PhosphatidylInositol-3-Kinase (PI3K)/AKT, MAPK/Extracellular signal-Regulated Kinases (ERKs), and WNT (an acronym of homologous wingless (Wg) and iNT-1 described in fly and mouse), facilitates metastasis [75]. Nowadays, the relevance of the CXCL12/CXCR4 axis in cancers has been ascertained through the impact of the CXCR4 genomic landscape in clinical settings. Data from approximately 19,000 patients genomically profiled across multiple cancer types at 8 academic medical centers showed that the genomic alteration rate (including mutation, copy number, and rearrangement) of the CXCR4 gene is 0.30%, accounting for 0.24% genetic loss, 0.03% CXCR4 non-sense (NS), 0.01% CXCR4 frame-shift (FS), and 0.01% CXCR4^R334X^ [76]. Thus, the characterization of the CXCR4 genomic landscape may represent a key and promising element in the identification of relevant biomarkers for prognosis, clinicopathological classification, and precision therapy in cancer.

## 2. CXCR4 Genomic Landscape in Tumors

### 2.1. Medulloblastoma

Although it has long been known that a pathologically activated CXC12/CXCR4 axis is significantly associated with increased proliferation or enhanced metastatic potential in a variety of human neoplasms [10,77,78], the first evidence of CXCR4 mutations dates back to 2005, when Schuller et al. first identified two mutations—one germline (A157C) and one somatic (C414T)—in human medulloblastoma, a frequent malignant brain tumor in childhood [79]. The A157C mutation was located in the first transmembrane region (TM1), which resulted in an amino acid change from isoleucine to leucine (I53L) within a domain that is highly conserved in evolution (Figure 2). The C414T mutation affected codon 97 through a substitution of negatively charged aspartate by neutral asparagine (D97N) in the second transmembrane region (TM2), a part of CXCR4 which is relatively close to the cell surface and involved in binding of both CXCL12 and the receptor antagonist plerixafor (AMD3100) [80]. This mutation has been shown in literature to both impair the inhibition of CXCR4 by plerixafor [81] and increase protein stabilization and surface expression of CXCR4 [82], suggesting its contribution to the pathological receptor activity or resistance to inhibitors. To date, further studies are still needed to establish the functional consequence of these mutations in medulloblastoma.

### 2.2. Colon Cancer and Melanoma Cell Lines

A second report of a spontaneously occurring CXCR4 mutation in human cancer cells was provided in 2009 by Ieranò et al., who detected a somatic point mutation (G574A) in cancer cell lines established from two metastatic human cancers: colon cancer (PD cell line) and melanoma (LB cell line) [83]. The mutation resulted in a valine to isoleucine amino acid substitution (V160L) in the fourth transmembrane region (TM4) of CXCR4 that is known to be involved in the binding of plerixafor [80] (Figure 2). Although the mutation does not affect in vitro receptor expression and migration in all cell lines, it appears to be detrimental to in vivo tumor growth. The observation that the mutation does not enhance cell growth is not surprising given the large number of other pathways that impact cell proliferation, but suggests that it may confer an advantage in terms of migration that may have been valuable during the process of tumor invasion and possibly metastases. Furthermore, the finding that treatment with plerixafor enhanced the growth of CXCR4-mutated cells suggests that this mutation may also affect the plerixafor binding site, such that upon binding to the receptor, plerixafor shifts its antagonist effects to agonist effects. 

### 2.3. Waldenstrom’s Macroglobulinaemia

In recent years, the onset of whole-genome sequencing has revealed CXCR4 as one of the most frequent somatic mutations identified in the indolent form of B-cell non-Hodgkin lymphomas (B-nHL), although their relevance for clinical presentation and overall survival, as well as their relationship with resistance to chemotherapy, are still unsolved issue. To date, it has been reported that approximately 30%–40% of patients with Waldenstrom’s macroglobulinemia (WM)—a type of lymphoplasmacytic lymphoma (LPL) exhibiting BM involvement and Immunoglobulin M IgM monoclonal gammopathy—carry mutations in heterozygosis of CXCR4 [84,85,86]. These somatic mutations are primarily subclonal, and almost always associated with MYeloid Differentiation primary response 88 (MYD88)^L265P^ mutation, the first identified recurring mutation in almost 67%–90% LPL/WM non-IgM secreting patients [87]. All 17 identified mutations result in both FS and NS mutations in the C-terminal region (C-ter) of the receptor (Figure 2) and are currently defined as WHIM-like mutations, as they closely resemble the already documented germline, activating mutations of CXCR4 C-ter occurring in heterozygosis in WHIM (warts, hypogammaglobulinemia, infection, and myelokathexis) syndrome (Online Mendelian Inheritance in Man (OMIM) 193670), a congenital autosomal dominant immunodeficiency disorder [88]. WHIM patients and their family members indeed carry several NS mutations (R334*, G336*, S338*, E343*), as well as FS (G323fs, L329fs, S339fs, S341fs) and missense (MS: E343K) mutations [89,90,91], which result in gain-of-function variants of CXCR4 [92] (see Appendix A, Figure 2). Recently, it has been reported that about 3.8% of all mutations occurring in the B-cell-receptor–CXCR4 signaling pathways identified in patients with follicular lymphoma (FL) affect the CXCR4 receptor [93]. However, unlike LPL/WM, only 50% of CXCR4 mutations in FL result in FS mutations of the CXCR4 C-ter that resemble WHIM-like mutations, whereas remaining mutations arise as MS mutations in regions distant from the C-ter (TM4 and TM5), similarly to those found by Ieranò et al. in their metastatic cell lines.

### 2.4. Other Cancers

Recently, several efforts have been made to integrate the amount of data coming from large-scale genome sequencing analysis of tumors, including the setting up of a dataset repository named The Cancer Genome Atlas (TCGA), which has collected and characterized more than 20 tumor types. Therefore, through a bioinformatics analysis of TCGA, it is possible to explore the tumor-associated genomic landscape of CXCR4. The analysis reveals the presence of several mutated variants of CXCR4, including WHIM-like mutations (Table 1). Nevertheless, the frequency of CXCR4 mutations is very low compared to LPL/WM (Figure 3). However, although an altered expression of CXCR4 has been reported for several cancer types (Figure 1), a significant frequency of CXCR4 mutated variants has been detected in only two cancer types—the uterine corpus endometrial carcinoma (uterine cancer, UCEC), which is the fourth most common cancer in females and accounts for approximately 80% of endometrial adenocarcinomas, and the DLBCL, the most common aggressive form of B-cell NHL (Table 1, Figure 2 and Figure 3). Importantly, while most of the CXCR4-mutated UCEC exhibit mutations in the N-terminal region (N-ter), variants in the C-ter have been reported in only one patient, where a WHIM-like mutation was identified. Conversely, in CXCR4-mutated DLBCL cases, both N-ter and C-ter mutations of CXCR4 exhibit almost the same frequency. In particular, the C-ter-mutated variants have been reported as chimeric proteins, where a region of CXCR4, including the C-ter, is replaced by CFLAR, an apoptosis regulator protein, along with Immunoglobulin Heavy Constant Alpha 2 (IGHA2), the constant region of immunoglobulin heavy chains.

Collectively, these observations strengthen the biological relevance of CXCR4 and the crucial impact of its genomic landscape in the pathogenesis of B-cell NHL malignancies. Moreover, similarly to LPL/WM, data in literature showed that a substantial proportion of activated B-cell-like subtype DLBCL (ABC DLBCL) also carry the MYD88^L265P^ mutation, whereas mutations in other small B-cell lymphomas are rare [87,96,97], suggesting that MYD88/CXCR4 interplay could represent a crucial hub in B-nHL malignancies.

## 3. CXCL12/CXCR4 Axis in Hematological Tumors: A Crucial Hub for B-Cell NHL Malignancies 

In hematological malignancies, as well as in solid tumors, the overexpression of CXCR4 is responsible for metastasis in organs expressing high CXCL12 levels (e.g., lymph nodes and BM), in addition to disease progression, increased tumor cell survival, and chemoresistance [98]. In chronic lymphocytic leukemia (CLL), the CXCL12/CXCR4 axis induces immune suppression via Signal Transducer and Activator of Transcription 3 (STAT3) phosphorylation in B and T-CLL cells [99]. Recently, p66Shc was identified as a negative regulator of CXCR4 endosome recycling in CLL [100]. High CXCR4 expression is found in B- and T-acute lymphoblastic leukemia (ALL) cells, where it correlates with higher incidence of relapse [70,101]. In B-ALL, chemotherapy upregulates CXCR4 expression in surviving leukemic blasts and accounts for the therapeutic resistance [102]. Recently, Notch signaling has been shown as a common feature in ALL, in particular in more than 60% of T-ALL [103,104,105]. Both hyperactive Notch1 and Notch3 enhances CXCR4 expression in thymus-derived and in BM-derived T-cells in T-ALL [106,107], and CXCR4 silencing inhibited the expansion of leukemic cells [106] through disruption of the Notch/CXCR4 partnership [102]. Of note, NFκB enhanced generation of Treg, suppressing the antitumor immune response in a Notch3-dependent T-ALL mouse model [108]. In AML, the activation of CXCR4 is critical for the migration and retention of leukemia cells within the BM, for extramedullary metastasis, chemotherapy resistance [109], and for the maintenance of minimal residual disease (MRD) [110]. Conversely, the inhibition of CXCR4 activity with motixafortide (BL-8040) in phase II clinical trials for relapsed or refractory T-ALL, lymphoblastic lymphoma, and AML induces the apoptosis of AML blasts [111]. Accordingly, CXCR4 is ascertained to be a crucial AML prognostic marker [112,113]. In fact, CXCR4 expression has prognostic value in 122 patients with cytogenetically normal AML and with an unmutated FLT3 gene. CXCR4 expression results in an independent factor associated with short overall survival (OS) and short event-free survival [113]. Du et al. showed that CXCR4 is a valuable prognostic marker in AML, in addition to age, leukocytosis, Fetal Liver Tyrosine Kinase 3 (FLT3) mutant, and extramedullary infiltration. High CXCR4 expression is also an independent risk factor for total survival time in patients with AML [114]. Finally, in multiple myeloma (MM) cells, CXCR4 expression correlated with disease progression [115], chemotherapy resistant MRD [116], and poor prognosis [117], while an increase in the CXCL12 serum expression level was associated with increased osteolytic disease and increased BM angiogenesis [118].

CXCR4 represents a crucial hub in B-cell biology. All subsets of B cells express CXCR4 throughout ontogeny, and genetic studies in knockout mice displayed severe defects in the generation of B cells, accounting for a key role of CXCR4 in regulating homeostasis of B cell compartments, as well as B-cell activation and migration [119]. Consistent with this, mice carrying CXCR4^WHIM-like^ mutations show defective B-cell development [120], and a minority of WHIM patients show increased susceptibility to Epstein–Barr virus (EBV)-related B-cell lymphoproliferative disorders and B-cell lymphomas [121], further demonstrating that CXCR4 signaling is essential for proper B-cell functionality. The most striking findings from the recent whole-genome sequencing have been the discovery of two exceptionally frequent activating somatic mutations affecting Toll-like/interleukin 1 (MYD88^L265P^) and CXCR4 receptor signaling in indolent (LPL/WM) and aggressive (DLBCL) forms of B-nHL [90], whereas other hematological tumors of different ontology, such as AML, were not affected (Figure 3). These observations suggest a crucial role of the CXCL12/CXCR4 axis in the pathogenesis and (potentially) the response to therapy and outcome of hematological tumors of B cell origin that nowadays comprise over 90% of the nHL worldwide, and which display great heterogeneity at the clinical, pathologic, and genetic levels, thus representing 4% of all new estimated cancer cases and deaths. The CXCR4 antagonist plerixafor was approved for the mobilization of hematopoietic stem cells for transplantation in patients with nHL or MM [122]. These advances have led to the recognition of CXCL12 and CXCR4 receptors as promising targets for cancer immunotherapy [3].

### 3.1. Waldenstrom’s Macroglobulinemia

WM is a rare form of nHL or lymphoplasmacytic lymphoma (LPL/WM) [91], with characteristic lymphoplasmacytic infiltration in the BM lymph nodes and spleen, along with the presence of monoclonal immunoglobulin M (IgM) protein in the blood [92]. The current therapeutic armamentarium relies on chemotherapy from nucleoside analogs to alkylating agents and proteasome inhibitors, and subsequently, the integration of rituximab, an anti-CD20 monoclonal antibody that now serves as the cornerstone of anti-LPL/WM regimens [93]. The approval of ibrutinib, the Bruton tyrosine kinase (BTK) inhibitor, both alone and in combination with rituximab, has expanded the treatment options for LPL/WM patients. LPL/WM remains an incurable entity and disease relapse is common. In LPL/WM, a point mutation of MYD88^L265P^ is the most common “gain-of-function” somatic mutation, occurring in 80%–95% of LPL/WM [123]. In LPL/WM cells, however, mutated MYD88 transcriptionally upregulates hematopoietic cell kinase (HCK), which in turn activates BTK, as well as the ERK and AKT [124], Phospholipase C (PLC-γ), and interleukin-1 and interleukin-4 receptor associated kinases (IRAK1 and IRAK4, respectively) [125]. Similarly, CXCR4 mutations closely resemble those observed in WHIM patients, and have been observed in 27%–40% of patients with LPL/WM [126]. The mutated CXCR4 receptor continues to generate pro-survival signals upon CXCL12 stimulation. The CXCR4^S338X^ clonality of ≥ 25% represents a potential biomarker for inferior responses to ibrutinib therapy [114]. Responses to ibrutinib and survival are not only impacted by MYD88^L265P^ but also by CXCR4 mutations [100]. A few drugs targeting CXCR4 are currently under investigation, as the response rates with ibrutinib alone are lower in the patient population with mutated CXCR4 compared to those who have the CXCR4 signature [127,128]. The CXCR4 antagonist ulocuplumab (BMS-936564/MDX1338) is a fully human IgG monoclonal antibody that is being evaluated in combination with ibrutinib in a phase 1/2 clinical trial for patients with relapsed or refractory WM (RRWM) only, who harbor a CXCR4 mutation (NCT03225716). Mavorixafor (AMD-070), an orally bioavailable agent with a half-life of approximately 23 h, is another noncompetitive, small molecule antagonist of CXCR4 currently under evaluation in combination with ibrutinib in RRWM (NCT 03995108) [129].

### 3.2. Diffuse Large B Cell Lymphoma

DLBCL is the most common type of non-Hodgkin lymphoma among adults. DLBCL typically presents as a nodal or extranodal mass with rapid growth. Overall survival (OS) has improved significantly since the new therapy based on rituximab with cyclophosphamide, vincristine, doxorubicin, and prednisone (CHOP) has been in use [130]. Nevertheless, refractoriness or relapse occur in one-third of patients, and most of them show poor long-term outcomes [131]. CXCR4 regulates tissue trophism of lymphoma cells and protects them from chemotherapy-induced apoptosis by promoting leukemic blasts homing into BM niches [132]. Lemma et al. [133] and Jahnke et al. [134] found that CXCR4 shows both nuclear and cytoplasmic distribution in lymphoma cells from primary central nervous system lymphoma samples. Higher nuclear expression is most strongly associated with DLBCL of nodal origin [132]. In 12 cases of primary testicular DLBCL [135] and 94 cases of DLBCL treated with rituximab-containing regimens [136], CXCR4 expression is associated with disease progression. However, in 20 patients with nHL, a significant decrease in CXCR4 mRNA levels in the BM after treatment correlates with a lower risk of death [137]. In a cohort of 743 patients with de novo DLBCL treated with standard rituximab–CHOP therapy, the expression of CXCR4 is associated with the male gender, tumor volume, ABC subtype, and high Ki-67 index, Myc, and Bcl-2 or p53 [138]. Another study showed poorer progression-free survival in training and validation cohorts of 468 and 275 subgroups, respectively, with germinal center B-cell-like (GCB) patients due to CXCR4 overexpression [138]. Peled et al. demonstrated that disrupting the CXCL12/CXCR4 axis using motixafortide is an effective way to abrogate the BM-derived mesenchymal stem-cell-mediated resistance of nHL cells to rituximab and to target nHL in the BM microenvironment [139]. Furthermore, plerixafor can enhance the effect of rituximab in DLBCL cell lines in vitro [140]. Previous studies have confirmed that abnormal activation of the PI3K/AKT/mTOR signaling pathway increases tumor proliferation and is associated with the poorer prognosis of patients with DLBCL [141]. The combination therapy with CXCR4 antagonist WZ811 and mTOR inhibitor Everolimus showed synergistic effects in DLBCL cell lines [142]. Furthermore, Kim et al. identified CXCR4 as a cause of primary resistance to idelalisib, a potent therapeutic agent that selective inhibits PI3Kδ in ABC DLBCL [143]. CXCR4 expression is a marker of DLBCL recurrence and is associated with shorter survival [136]. Primary refractory cells expressed high levels of CXCR4 on the cell surface and treatment with CXCR4 antagonists combined with idelalisib resulted in a profound synergistic effect of growth inhibition compared to individual treatment [143].

## 4. CXCR4 WHIM-Like Mutations in LPL/WM 

The most striking findings from recent whole-genome sequencing in LPL/WM have been the seminal discoveries of MYD88^L265P^ mutation, which are present in the vast majority of cases (85%–100%), and CXCR4 mutations, identified in nearly one-third of patients (who almost exclusively harbor the MYD88^L265P^ variant). These discoveries have laid a solid foundation for a paradigm shift in the diagnostic and therapeutic approaches towards this rare hematologic malignancy [144]. Unfortunately, a paucity of knowledge regarding the CXCR4 genomic landscape in cancer exists because of the low incidence of MYD88 mutations in other B-cell diseases, and because CXCR4 somatic mutations have not been previously described in any malignant condition [145].

Hunter and Treon et al. first analyzed the CXCR4 gene in CD19-positive selected lymphocytes in LPL/WM cases and showed that 30%–40% of WM patients carry activating somatic, subclonal mutations in the receptor C-ter, which are almost always associated with MYD88^L265P^ mutations [85,86]. As previously described, the locations of these mutations in the C-ter are similar to those observed in WHIM syndrome (Appendix A, Figure 2). Particularly, two classes of CXCR4 mutations have been reported, including both NS (CXCR4^WHIM/NS^) mutations that truncate the distal 15- to 20-amino acid region, and FS (CXCR4^WHIM/FS^) mutations that comprise a region of up to 40 amino acids in the C-ter. NS and FS mutations are almost equally distributed among LPL/WM patients. Several preclinical studies have demonstrated that these mutations reproduce the effects of CXCR4 overexpression in LPL/WM cells, leading to increased disease dissemination in vivo and enhanced adhesion properties and growth in vitro, thus acting as driver mutations of tumor progression and modulators of drug resistance. Indeed, the most common CXCR4 WHIM-like NS mutation (CXCR4^WHIM/NSS338X^) identified in LPL/WM has shown enhanced and sustained AKT and ERK activation and impaired pro-apoptotic signature upon CXCL12 stimulation, progressing into increased cell migration, adhesion, growth, and survival of LPL/WM cells [146]. Accordingly, mutated LPL/WM cells also display an enrichment for mRNAs for genes involved in MAPK, BTK, and PI3K pathways related to oncogenesis, invasiveness, proliferation, and anti-apoptosis, thus further confirming the activating role of the CXCR4^WHIM^ variants in LPL/WM [147]. Furthermore, several reports demonstrated that these mutations have a crucial impact on responses to therapeutic treatments. Indeed, gene set enrichment analysis showed that genes related to drug resistance were enriched in CXCR4^WHIM^ LPL/WM cells, whereas genes related to drug responsiveness were enriched in control cells. In particular, mutated LPL/WM cells displayed resistance to the suppressive effects of BTK-based therapy on AKT and ERK signaling, as well as PI3K and mTOR inhibitors, whereas the response to proteasome inhibitors (i.e., bortezomib) was unaffected [146,147,148,149,150]. Therefore, the key findings that emerged from these studies are that the mutational status of MYD88 and CXCR4 identify distinct subsets of patients and seem to have dramatic impacts on clinical presentation and treatment response.

Translating the genomic landscape of CXCR4 from the bench-to-bed side, several reports correlate CXCR4^WHIM^ variants to clinicopathological presentation of LPL/WM patients, suggesting a crucial contribution of CXCR4 mutational status in the diagnosis. Indeed, CXCR4^WHIM^ LPL/WM cases identify a subgroup of LPL/WM patients with higher disease activity, showing more aggressive disease features at presentation, with higher serum IgM levels and more CD20-positive lymphoplasmacytic BM infiltration but lower levels of leucocytes, hemoglobin, and platelets. These patients also show a symptomatic status, including hyperviscosity syndrome, requiring therapy with bortezomib, as indicated by National Comprehensive Cancer Network guidelines [151]. However, recent data indicate that this might be mainly restricted to cases carrying CXCR4^WHIM/NS^ rather than CXCR4^WHIM/FS^ mutations. Indeed, patients with MYD88^L265P^CXCR4^WHIM/NS^ show higher BM involvement and IgM serum levels, and are more inclined to have symptomatic disease requiring therapy at presentation [152]. Conversely, patients with MYD88^WT^CXCR4^WT^ manifest lower BM involvement, whereas patients with either MYD88^L265P^CXCR4^WT^ or MYD88^L265P^CXCR4^WHIM/FS^ show intermediate levels of BM disease involvement. Patients with CXCR4^WHIM^, regardless of whether they carry NS or FS mutations, are also less likely to manifest lymphadenopathy. According to the increased signaling activity of CXCR4^WHIM^, these findings suggest a more pronounced tropism of the BM for LPL/WM cells carrying WHIM-like mutations, and account for cell survival and IgM release. Moreover, this evidences clearly indicates that NS and FS mutations behave differently. In WHIM syndrome, very few patients carry FS mutations, whereas NS mutations have been identified in almost all patients [92,153]. Unfortunately, currently no preclinical modeling has been done to understand the activating nature of FS mutations.

Despite the aggressive clinical impact of CXCR4^WHIM/NS^ mutations, overall survival for these patients is not adversely impacted. Analogue frequency of deaths occurred in both CXCR4^WT^ (6%) and CXCR4^WHIM^ (4%) patients who harbored the MYD88^L265P^ mutation [85]. However, a germline polymorphism in CXCL12 (-801GG) at RS1801157 has been identified as a new genetic adverse prognostic factor for all LPL/WM patients associated with shorter median survival after initiation of first line therapy [154]. Increased expression of CXCL12 (−801GA) polymorphism might also contribute to retaining LPL/WM cells within their protective niche in the BM in close and prolonged contact with stromal cells, where they receive additional signals that may be implicated in cell proliferation or survival, as observed during the development of normal B lymphocytes in the BM or in the lymph node [155]. This finding strengthens the clinical significance of the genetic background of the CXCL12/CXCR4 axis in LPL/WM, and prospective studies incorporating CXCL12 polymorphisms and CXCR4 mutation status in assessing treatment response, progression-free survival, and overall survival in LPL/WM patients could nonetheless clarify the predictive and prognostic roles of CXCL12 and CXCR4 variants, both alone and together. Unfortunately, to date the revised International Prognostic Scoring System for Waldenström Macroglobulinemia (IPSSWM) does not include the mutational status of either MYD88 or CXCR4 [156], as it is not routinely performed and is available in only a few patients. It has been associated with prognosis in some studies [85] but not confirmed in others [123], and it has been considered as predictive for response to ibrutinib [157,158]. However, additional follow up is needed to confirm the prognostic impact, especially in non-ibrutinib-treated patients [159]. 

From a clinical standpoint, the potential to translate the findings of MYD88 and CXCR4 mutations into therapeutic gains for LPL/WM patients is noteworthy. Patients with MYD88^L265P^/CXCR4^WT^ show the highest rates of response, whereas a reduced response rate is demonstrated for patients harboring MYD88^L265P^/CXCR4^WHIM^. These observations in clinical settings recapitulate data obtained in in vitro studies where CXCR4^WHIM/S338X^-engineered LPL/WM cells showed a transcriptome signature enriched for drug-resistance gene expression and were resistant to ibrutinib treatment [157]. As such, the optimal therapeutic strategies to extend clinical remission in patients who have become refractory to ibrutinib are unclear and remain an active area of investigation. However, these studies highlight the importance of understanding both MYD88 and CXCR4 mutation statuses in LPL/WM and may provide the basis for a more personalized treatment approach. Nowadays, targeting CXCR4 represents an innovative, interesting therapeutic approach in patients with WM, either harboring or not harboring the CXCR4^WHIM^ mutation [160]. In addition to plerixafor, several other antagonists to CXCR4 have been developed and are in clinical trials, including uloclupumab, mavorixafor, and burixafor (TG-0054), and could be investigated for use either alone or in combination treatment strategies for LPL/WM patients with CXCR4 mutations. Interestingly, it has been recently reported that the novel anti-CXCR4 antibody ulocuplumab shows an anti-LPL/WM activity by inhibiting tumor growth both in in vitro and in vivo studies [147]. Ulocuplumab is a fully humanized anti-CXCR4 monoclonal antibody with a longer half-life than plerixafor, which shows pro-apoptotic activity in the context of hematological malignancies [161]. In LPL/WM, ulocuplumab-dependent CXCR4 neutralization shows equal efficacy in targeting both CXCR4^WT^ and CXCR4^WHIM^ LPL/WM cells through inhibition of AKT, ERK, and SRC phosphorylation, as well as increased GSK3β and β-catenin phosphorylation, leading to β-catenin degradation [147]. Moreover, ulocuplumab triggers apoptosis by activation of caspase-9 and Poly (ADP-ribose) polymerase (PARP) cleavage [147]. Overall, these observations lay the groundwork for clinical trials testing ulocuplumab for treatment of LPL/WM patients either harboring or not harboring the CXCR4^WHIM^ mutation.

## 5. CXCR4 Targeting in Hematological Tumors

CXCR4 is considered as one of the best potential targets in hematological tumors. Several small molecules, peptides, and antibodies targeting the CXCR4 axis, such as plerixafor, BL-8040/BKT140, LY2510924, PF-06747143, BMS-936564, and NOX-A12, have shown promising results in leukemia, lymphoma, and myeloma [98]. Several ongoing clinical trials have proposed CXCR4 inhibitors for treatment of hematological tumors alone or in combination with chemotherapy or biological agents (Figure 4). Plerixafor was originally modeled after a predecessor called JM1657, which had been identified as an impurity in a commercial (mono)cyclam preparation intended as an anti-HIV agents and which selectively blocked the CXCR4 receptor. [162]. Other specific CXCR4/CXCL12 antagonists have been developed and are in different phases of clinical development. Mavorixafor (also called AMD11070) is a small molecule that is a selective and orally bioavailable antagonist of CXCR4 [163] in Phase 1b/2a development in association with immune checkpoint inhibitors in melanoma (NCT02823405) and renal cancer (NCT02923531). Balixafortide (POL6326) is a CXCR4 antagonist in the form of cyclic peptide that effectively mobilizes hematopoietic stem and progenitor cells in healthy volunteers [164]. The objective response of balixafortide plus eribulin reached 30% (16/54) in the treatment of metastatic breast cancers [165]. Balixafortide treatment versus eribulin is currently being evaluated in a phase 3 trial (NCT03786094). Motixafortide (BL-8040, 4F-benzoyl-TN14003) is a 14-amino acid peptide antagonist against CXCR4. This peptide could stimulate the recovery of BM after transplantation [166] and induce the apoptosis of human acute myeloid leukemia blasts [166]. A phase 3 trial (NCT03246529) is currently evaluating the use of motixafortide for stem cell mobilization [167]. Actually, there are two fully human anti-CXCR4 antibodies—PF-06747143 [116] and ulocuplumab [168]—used for the treatment of hematologic malignancies. Finally, LY2510924, is a potent cyclic peptide antagonist of CXCR4 with acceptable in vivo stability and a pharmacokinetic profile similar to a typical small molecule inhibitor [169], which is being evaluated in phase 1/2 clinical trials in extensive small cell lung cancer (NCT01439568), renal cancer (NCT 01391130), and in association with durvalumab (an anti Programmed Death-Ligand 1 (PD-L1) in solid tumors (NCT02737072). Finally, there is one CXCL12 antagonist, olaptesed pegol (NOX-A12), a pegylated structured L-oligoribonucleotide that binds and neutralizes CXCL12, which is in a phase 2a study of relapsed and refractory lymphocytic leukemia (NCT01486797) [132]. However, limitations and challenges related to CXCR4 should be underlined. CXCR4 is ubiquitously expressed in healthy normal cells; thus, CXCR4 signaling can produce several side effects due to the interference with immunological and physiological responses. Nevertheless, plerixafor and other CXCR4 antagonists [162] have shown minimal side effects in clinical studies. Small molecules and small peptides that target the CXCL12/CXCR4 axis pose several challenges in MM treatment. this is due to the unwanted side effects of the small molecules due to ubiquitous CXCR4 expression in different sites, with the latter often showing poor selectivity [170]. Potential mechanisms of resistance include down-regulation or internalization of surface CXCR4, heterogeneous expression in cancer cells resulting in incomplete targeting, and potential competition with locally increased CXCL12 concentrations [25]. Therefore, the current challenge to overcome in clinical translation is to identify powerful and specific CXCR4 modulators to be optimized for treatment schedules and drug synergism. 

## 6. Conclusions

Thus, the characterization of the genomic landscape of CXCR4 represents a key and promising element in the identification of relevant biomarkers for clinicopathological classification, prognosis, and precision therapy in LPL/WM, providing a crucial contribution for the explanation of the molecular mechanisms underlying the biological heterogeneity of this tumor, which still remains an unsolved issue. Unfortunately, the analysis of the genomic profile of CXCR4 is not routinely performed in clinics for LPL/WM patients. Importantly, the sensitivity of the used sequencing technique (allele-specific polymerase chain, Sanger, or whole-genome sequencing) and the nature of the analyzed sample (CD19-positive selected lymphocytes or whole BM) represent the most important issues in defining the mutational status of CXCR4 [152,171]. Therefore, universal guidelines are required to standardize the analysis and bring CXCR4 genomic profiling into clinics. Interestingly, Bagratuni et al. identified peripheral blood cell-free DNA as a useful, minimally invasive, cost-effective, and time-effective tool for the identification of the presence of both MYD88 and CXCR4 mutations in patients with IgM monoclonal gammopathies, avoiding unnecessary BM assessment [172].

## Figures and Tables

**Figure 1 vaccines-08-00164-f001:**
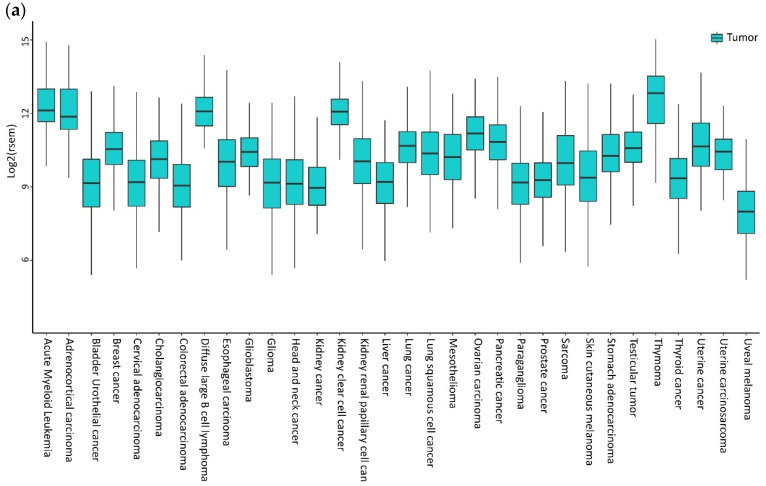
CXCR4 expression in tumors. (**a**) CXCR4 expression in different tumors is investigated by public gene expression data from the Firebrowse repository [71]. Gene expression levels (as normalized RNA-Seq by Expectation Maximization (RSEM) values) were downloaded and distribution boxplots were performed using the ggplot2 R package (version 2_3.2.0) [72] (**b**) Differential expression analysis between normal and tumor samples was performed using the edgeR package (R version 3.5.1; edgeR version 3.24.3) [73,74]. *p*-values are defined with Fisher’s exact Test.

**Figure 2 vaccines-08-00164-f002:**
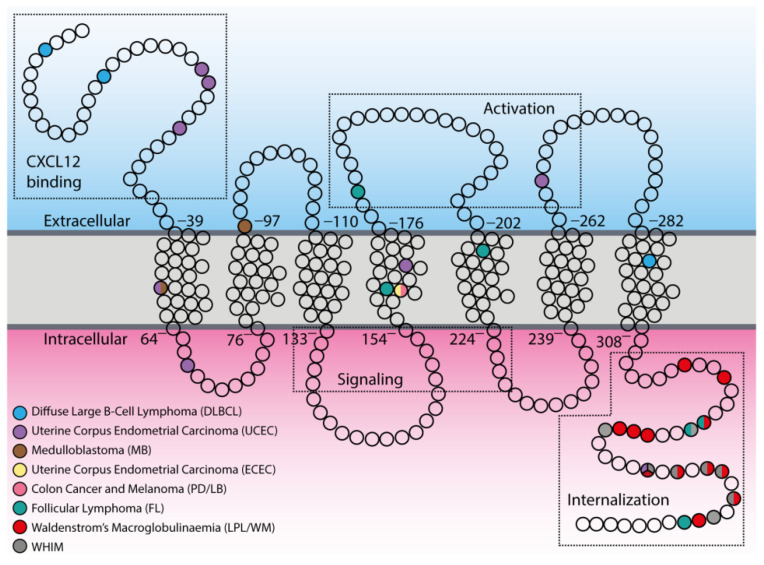
Distribution of tumor-associated mutations within CXCR4 topology. In the scheme, principal components of the CXCR4 structure are indicated with the dotted boxes in both extracellular and intracellular compartments (CXCL12 binding, activation sites, signaling, and internalization regions). Empty circles represent single amino acids, while colored circles indicate mutated ones.

**Figure 3 vaccines-08-00164-f003:**
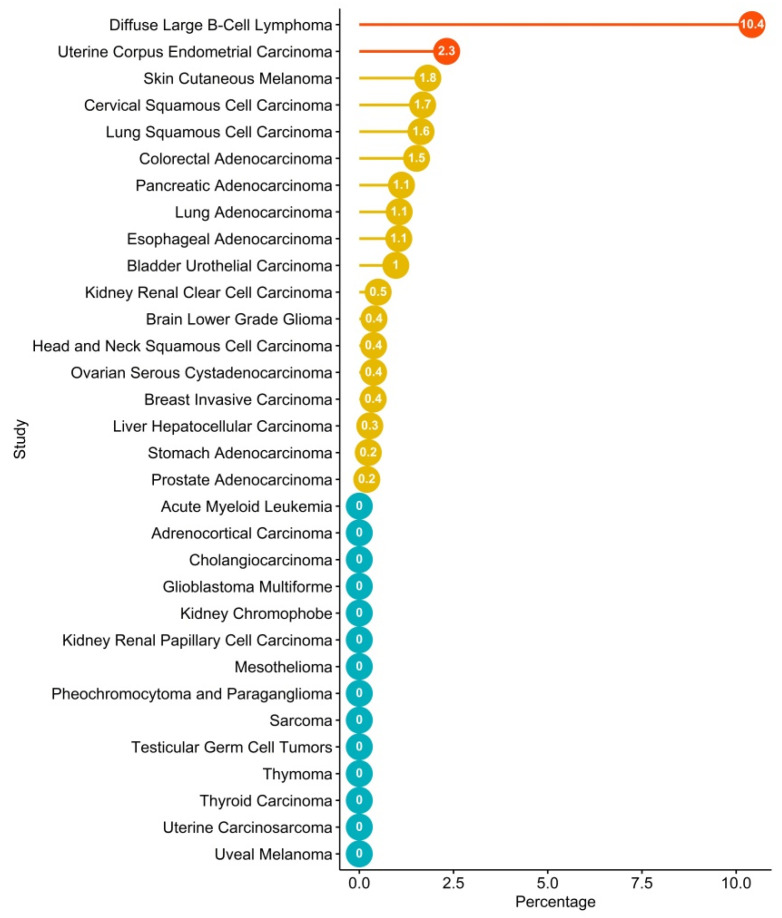
TCGA screening for frequency of CXCR4 mutations in tumors. TCGA patients’ mutations were investigated in cBioPortal [94], taking into consideration ones associated with CXCR4. Bar plot of frequency of CXCR4 expression in different tumors according to the TCGA dataset. Red and yellow dots refer to tumors that are statistically significant or non-significant, respectively, according to Fisher’s exact test. Blue dots refer to tumors where CXCR4 mutations are not identified. Plot was performed through Rstudio with ggplot2 [72] and ggpubr libraries [95].

**Figure 4 vaccines-08-00164-f004:**
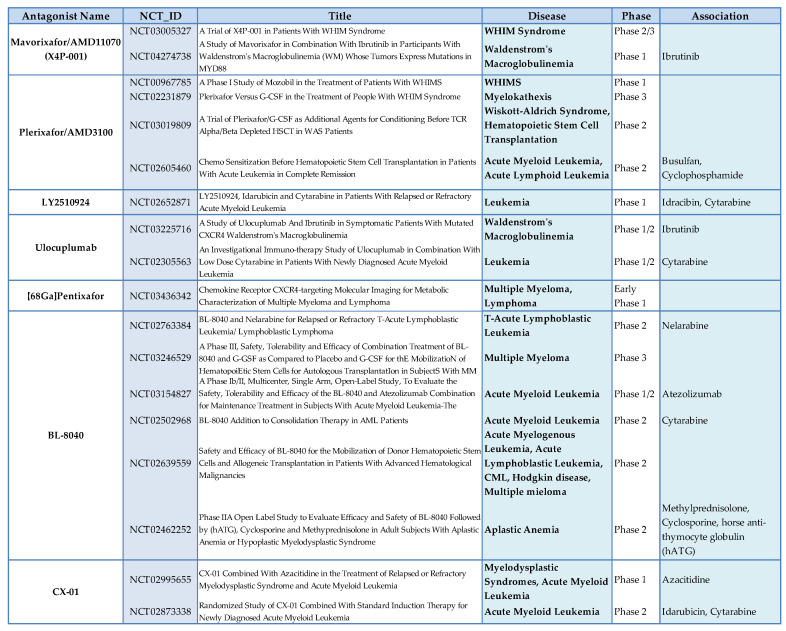
CXCR4 antagonists in hematological tumors.

**Table 1 vaccines-08-00164-t001:** The Cancer Genome Atlas (TCGA) screening for CXCR4 mutations in tumors.

Tumor	CXCR4 Mutations ^1^	*p*-Value ^2^
Diffuse Large B-Cell Lymphoma	**CXCR4-CFLAR (F)** - **CXCR4-IGHA2 (F)** - A291T (MM) – G17D (MM) - I4S (MM)	0.00016
Uterine Corpus Endometrial Carcinoma	**R334 * (NM)** - I265V (MM) - L167M (MM) - K67N (MM) - I53V (MM) - R30C (MM) - E26K (MM) - K25N (MM) – **X6_splice (SR)**	0.015
Skin cutaneous melanoma	**G336R (MM)** - L165F (MM) - P163L (MM) - P92S (MM) - K75M (MM) - G19E (MM) - M16I (MM) - S9L (MM)	0.08
Cervical adenocarcinoma	**V340G (MM)** - R334 * (NM) - L238F (MM) - I185V (MM) - E14K (MM)	0.2
Lung squamous cell cancer	**E345D (MM)** - **V320Cfs *24 (FSI)** - H281R (MM) - L165I (MM) - R148K (MM) - R134L (MM) - Y76C (MM) - M24I (MM)	0.14
Colorectal Adenocarcinoma	**R334 * (NM)** - **L329I (MM)** - R235C (MM) - D181N (MM) - R134H (MM) - H113R (MM) - G19W (MM) - A250T (MM) - **S18 * (FSD)**	0.19
Pancreatic cancer	S339Y (MM) - S263F (MM)	0.7
Lung cancer	**S347L (MM)** - **E345Q (MM)** - T287N (MM) - E153K (MM) - R77K (MM) - G3W (MM)	0.6
Esophageal carcinoma	I162M (MM)	0.6
Bladder Urothelial cancer	**S312A (MM)** - **K310E (MM)** - K239R (MM) - R183K (MM)	0.8
Kidney clear cell cancer	I270V (MM) - I243T (MM)	0.6
Brain Lower Grade Glioma	**E343G (MM)** - R30C (MM)	0.3
Head and Neck cancer	V198L (MM) - V112L (MM)	0.3
Ovarian carcinoma	R235P (MM) - Y76N (MM)	0.3
Breast cancer	**T311Pfs *10 (FSD)** - Q202K (MM) - S144R (MM) - L125F (MM)	0.08
Liver cancer	I222F (MM)	0.4
Stomach adenocarcinoma	**S351Y (MM)**	0.19
Prostate cancer	**ARGLU1-CXCR4 (F)**	0.13

^1^ TCGA patients’ mutations were investigated in cBioPortal [94]. Bold refers to mutations affecting CXCR4 C-ter. NM: nonsense mutation (*: stop codon); MM: missense mutation; F: fusion. FSI: frameshift insertion; FSD: frameshift deletion. ^2^
*p*-values are defined with Fisher’s exact Test.

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
