# Peer review of "New Insights on the Emerging Genomic Landscape of CXCR4 in Cancer: A Lesson from WHIM"

_vaccines, 2020, doi:10.3390/vaccines8020164_

Round 1

Reviewer 1 Report

In this manuscript "New insights on emerging genomic landscape of CXCR4 in cancer: a lesson from WHIM” Stefania Scala et.al., summarized the current knowledge on the consequence of CXCR4 mutations in cancer biology, focusing on its importance as predictors of clinical presentation and response to therapy.

The comments and suggestions for this manuscript are as follows-

  1. The authors should provide a more comprehensive introduction and conclusion with some more recent published data and appropriate references.
  2. Page 4, figure1. The author must increase the text font on X and Y axis legends. In the current format, legends are not legible.
  3. Page 7, Table 1. The author is advised to add the reference after the p-value column, otherwise, the tables are concise including proper statistical analysis, wherever required.

Author Response

We thank Reviewer #1 for the comments. We agree with all of them and we apologize for the lack of clarity on several aspects.

  1. The authors should provide a more comprehensive introduction and conclusion with some more recent published data and appropriate references. Response: we thank Reviewer #1 for the comments. In order to provide a more comprehensive introduction and conclusion with more recent and appropriate references, we made the following main changes to the text: (1) we provide a more extended version of Paragraph 1.1; (2) we provide a more detailed description of CXCR7 contribution to CXCL12/CXCR4 axis in tumor biology in Paragraph 1.2. Moreover, several minor changes have been provided within the text.
  2. Page 4, figure1. The author must increase the text font on X and Y axis legends. In the current format, legends are not legible. Response: as indicated by reviewer, we increased the text font on X and Y axis legends. We provide a new modified version of Table 3.
  3. Page 7, Table 1. The author is advised to add the reference after the p-value column, otherwise, the tables are concise including proper statistical analysis, wherever required. Response: The Cancer Genome Atlas is a database collecting over that 10 thousands of sequenced samples data, coming from a several amount of tumor types. Moreover, it collected many types of data for each tumor and normal samples, such as clinical information (e.g. smoking status), molecular analyte metadata (e.g. sample portion weight) and molecular characterization data (e.g. gene expression values). All this information is included in cBioportal, a web tool developed by TCGA, from which researchers can download data and analyze them. Therefore, since information about mutations are provided by TCGA and they are not coming from literature, it is not possible to add the reference after the p-value column as suggested by reviewer. We hope the explanation satisfied the reviewer’s request.

Reviewer 2 Report

The manuscript is clear, concise, and mostly well written. there are a few minor typos that need to be correct. I don't have a lot more to add concerning this manuscript. It is an excellent review article and should be published. It covers the known data on CXCL12 and CXCR4 well and I like how the manuscript is inclusive foe a very wide number of different human malignancies. I small point is that I would like to see a little more about therapies that target these genes products in cancer treatment.  This part of the manuscript seems a little thin. However, overall it is an excellent review and after a few typos are addressed, it should be published.

As a note my review took longer than usual as COVID19 testing implementation took precedence over all other work activities.

Author Response

We thank Reviewer #2 for the comments that helped improve the clarity and focus of this review. In agreement with his/her last statement we provided an extended description of the current therapies that target CXCR4/CXCL12 in cancer treatment in Paragraph 3 and mainly in Paragraph 6.